# EquivaMap: Leveraging LLMs for Automatic Equivalence Checking of Optimization Formulations

**Haotian Zhai** [1]   **Connor Lawless** [2]   **Ellen Vitercik** [2]   **Liu Leqi** [1]

## Abstract

A fundamental problem in combinatorial optimization is identifying equivalent formulations. Despite the growing need for automated equivalence checks—driven, for example, by *optimization copilots*, which generate problem formulations from natural language descriptions—current approaches rely on simple heuristics that fail to reliably check formulation equivalence. Inspired by Karp reductions, in this work we introduce *Quasi-Karp equivalence*, a formal criterion for determining when two optimization formulations are equivalent based on the existence of a mapping between their decision variables. We propose *EquivaMap*, a framework that leverages large language models to automatically discover such mappings for scalable, reliable equivalence checking, with a verification stage that ensures mapped solutions preserve feasibility and optimality without additional solver calls. To evaluate our approach, we construct *EquivaFormulation*, the first open-source dataset of equivalent optimization formulations, generated by applying transformations such as adding slack variables or valid inequalities to existing formulations. Empirically, *EquivaMap* significantly outperforms existing methods, achieving substantial improvements in correctly identifying formulation equivalence.[1]

## 1. Introduction

Combinatorial optimization lies at the heart of many of today's most pressing challenges in operations research,
theoretical computer science, and machine learning. Its applications range from classic problems such as shortest path (Korte & Vygen, 2012) and maximum flow (Schrijver, 1983) to modern challenges in neural architecture search (Elsken et al., 2019) and hyperparameter optimization (Khadka et al., 2024).

A fundamental problem in combinatorial optimization is identifying equivalent formulations. Historically, establishing equivalence has played a pivotal role in unifying problem-solving techniques and advancing theoretical characterizations of a problem's computational complexity. In theoretical computer science, equivalence between problems underpins the concept of NP-completeness (Cook, 1971; Karp, 1972), which unifies many seemingly distinct problems—such as SAT, Vertex Cover, and Subset Sum—into the same equivalence class. This unification enables researchers to prioritize the development of algorithms for canonical problems while ensuring their applicability across equivalent problems. Similarly, in applied fields such as network design (Kan, 1978) and semiconductor scheduling (Fang et al., 2023), recognizing equivalence between optimization problems has historically facilitated the transfer of algorithms, reducing duplication of effort.

The advent of large language models (LLMs) has exposed a new frontier in combinatorial optimization, introducing opportunities to automate problem formulation, while also presenting new challenges—chief among them, the need for reliable equivalence checking. Recent research has focused on developing *optimization copilots*, systems that automate the translation of natural language descriptions into formal optimization formulations, particularly for mixed-integer linear programming (MILP) problems (Ramamonjison et al., 2023; Xiao et al., 2023; AhmadiTeshnizi et al., 2024; Astorga et al., 2024). These advancements hold significant potential for democratizing access to optimization techniques, broadening the reach of powerful tools for better decision-making (Wasserkrug et al., 2024). However, the widespread adoption of optimization copilots hinges on reliable evaluation mechanisms capable of verifying whether the generated formulations are equivalent to their ground-truth counterparts. Moreover, automatic formulation equivalence checking is critical to improving optimization copilots by serving

[1]The University of Texas at Austin, TX, USA [2]Stanford University, CA, USA. Correspondence to: Haotian Zhai <haotian.zhai@utexas.edu>.

*Proceedings of the $42^{nd}$ International Conference on Machine Learning*, Vancouver, Canada. PMLR 267, 2025. Copyright 2025 by the author(s).

[1]The code and datasets are available at https://github.com/HumainLab/EquivaMap and https://huggingface.co/datasets/humainlab/EquivaFormulation.

as an intermediate step, facilitating more efficient formulation search and refinement (Astorga et al., 2024).

Despite the importance of equivalence checking in combinatorial optimization, existing automatic approaches rely heavily on heuristics (e.g., comparing optimal objective values) and lack a precise, universally accepted definition of what constitutes formulation equivalence. Formal methods such as Karp reductions (Karp, 1972) offer valuable theoretical insights into problem equivalence but were not designed for modern automated settings, often requiring considerable human time and expertise to construct.

Towards precise and reliable equivalence checking, we propose a formal definition of formulation equivalence—*Quasi-Karp Equivalence*—grounded in the principles of Karp reductions. Quasi-Karp Equivalence determines whether two formulations are equivalent by checking for the existence of a mapping between their decision variables. We propose *EquivaMap*, an approach that automates equivalence checking by using LLMs to identify mappings between formulations, followed by a lightweight verification step to ensure these mappings preserve optimality and feasibility without additional MILP solver calls. Grounded in a precise definition of formulation equivalence, *EquivaMap* allows for automatic equivalence verification for optimization formulations.

Our contributions can be summarized as follows:

- We identify pitfalls of existing equivalence checking methods (Section 3.1).

- We propose Quasi-Karp equivalence as a formalism for defining when two optimization formulations are equivalent through the existence of a mapping between their decision variables (Section 3.2) and present *EquivaMap*, a scalable method that uses LLMs to discover candidate mappings, paired with a separate verification step to ensure correctness (Section 3.3).

- To evaluate the performance of equivalence-checking methods, we introduce, to the best of our knowledge, the first dataset—*EquivaFormulation*—that documents both equivalent formulations and the transformation between them (Section 4.1). Empirically, we show that *EquivaMap* outperforms existing methods across various equivalent transformations (Section 4.2).

## 2. Background and Related Work

Our work connects important lines of research on combinatorial optimization (especially MILPs), LLMs for MILP modeling, and automatic equivalence-checking methods for optimization formulations.

### 2.1. Combinatorial Optimization and MILPs

Combinatorial optimization (CO) broadly deals with finding an optimal object from a finite (or countably infinite) set of feasible candidates. Such problems arise in diverse fields, including operations research, computer science, and engineering, where discrete variables model decisions in practical scenarios such as routing, scheduling, or allocation of limited resources (Papadimitriou & Steiglitz, 1998).

A foundational tool for combinatorial optimization is *mixed-integer linear programming* (MILP), formulated as:

$$\min_{x \in \mathbb{R}^p \times \mathbb{Z}^{n-p}} c^\top x \tag{1}$$
$$\text{subject to} \quad Ax \circ b, \quad \ell \leq x \leq u,$$

where $x$ is the vector of decision variables, $c$ is the cost vector, $A$ is the constraint coefficient matrix, and $b$ is the vector of constraint bounds. The notation $Ax \circ b$ represents a system of linear constraints, where $\circ$ denotes relational operators from the set $\{\leq, \geq, =\}$. The variables $x$ are partitioned into $p$ continuous variables and $n-p$ integer variables. Let $x^*$ denote an optimal solution to (1), and let $z^* = c^\top x^*$ be the corresponding optimal objective value. If all decision variables are continuous ($p = n$), the problem is a *linear program* (LP). MILPs capture many prominent combinatorial problems such as the traveling salesman problem (TSP) (Cook et al., 2011), knapsack problem (Pisinger & Toth, 1998), and network design problems (Kan, 1978).

Many fundamental CO problems—including TSP and Knapsack—are known to be NP-hard. A key contribution to the theory of NP-completeness was provided by Karp (1972), who demonstrated that a number of widely studied problems are mutually reducible in polynomial time (often referred to as "Karp reductions"). These reductions establish deep structural connections among CO problems, showing that if a polynomial-time algorithm exists for one, it can be systematically adapted to solve many others.

### 2.2. Language Models for MILP Modeling

The use of language models for MILP modeling has sparked considerable interest in the AI-for-OR community. The NL4Opt competition (Ramamonjison et al., 2023) focused on using natural language processing (NLP) methods to formulate optimization problems based on their text descriptions. More recently, with the advent of LLMs, a number of LLM-based *optimization copilots* aim to automate MILP modeling (Mostajabdaveh et al., 2024; Ahmed & Choudhury, 2024; Li et al., 2023b; Yu & Liu, 2024; Huang et al., 2024a; Kadıoğlu et al., 2024; Yang et al., 2024). Both the Chain-of-Experts (Xiao et al., 2023) and OptiMUS (AhmadiTeshnizi et al., 2024) frameworks designed LLM-based multi-agent systems to automate the modeling of complex optimization problems by leveraging the reasoning capabili-

ties of the LLMs. Tang et al. (2024) further demonstrated the potential of LLMs by fine-tuning open-source models with synthetic data tailored for modeling optimization problems, achieving significant performance improvements over baseline methods. Building on these capabilities, LLM-powered chatbots have been used to allow users to interact with optimization models in a number of contexts including supply chain management (Li et al., 2023a), meeting scheduling (Lawless et al., 2024b), debugging infeasible models (Chen et al., 2023), and improving solver configurations (Lawless et al., 2024a). These advancements highlight why LLMs are particularly suitable for MILP modeling: their ability to process and generate structured information from natural language aligns well with the requirements of optimization problem formulation. The rapid development of optimization copilots underscores the need for reliable, scalable evaluation techniques.

### 2.3. Existing Automatic Equivalence Checking Methods

The central task of evaluating optimization copilots is automatically checking whether the generated formulation is equivalent to a ground-truth correct one. The earliest method used in the NL4OPT benchmark (Ramamonjison et al., 2023) for evaluating formulation equivalence is *canonical accuracy*, which looks at direct equivalence between declarations (e.g., objective, constraints) between a reference correct formulation and a generated formulation. This method is sensitive to permutations of the order of the declarations in a formulation and fails when multiple valid formulations exist for the same problem. The method used in benchmarks such as NLP4LP (AhmadiTeshnizi et al., 2024), MAMO (Huang et al., 2024b), and IndustryOR (Tang et al., 2024) is *execution accuracy*, which evaluates whether two MILP formulations are equivalent by solving them (using a MILP solver such as Gurobi) and checking if they have the same optimal objective value. Execution accuracy is sensitive to variable re-scaling, which can create inconsistencies even when the formulations are functionally equivalent. To address these issues, recent approaches utilize Graph Edit Distance (Xing et al., 2024) and a modified Weisfeiler-Lehman (WL) test (Wang et al., 2024) to measure structural similarity between the generated and reference formulations. These methods offer insights into equivalence beyond the optimal objective value but have limitations. They are particularly sensitive to structural modifications, such as adding cutting planes, which keep the formulation equivalent but change its structural information. Beyond these methods, Steever et al. (2022) proposed an image-based approach to detect structural similarity among large-scale MILPs.

## 3. Methodology

This section introduces *Quasi-Karp Equivalance* and *EquivaMap*, our method for leveraging LLMs to automati-

cally check such equivalence. In the general setup, we have two formulations $\alpha$ and $\alpha'$ corresponding to the same (feasible) optimization problem $\mathcal{P}$, with optimal objective values $z^*$ and $z'^*$ respectively. For example, Figure 1 presents two formulations $\alpha$ and $\alpha'$ of an optimization problem $\mathcal{P}$ — the *stable set problem*. Our method aims to evaluate the equivalence of $\alpha$ and $\alpha'$ for a given *instantiation* of the problem. In Figure 1, an instantiation of $\mathcal{P}$ would be defined by a specific input graph.

### 3.1. Pitfalls of Existing Equivalence Checking Methods

We discuss existing methods for evaluating formulation equivalence, including canonical accuracy, execution accuracy, and the WL-test, and exhibit settings where these methods fail.

**Canonical accuracy** is based on matching declarations between predicted and reference programs, where a declaration represents either an optimization objective or a constraint (Ramamonjison et al., 2023).

**Definition 3.1** (Canonical Accuracy). Given a reference declaration $d$ (objective or constraint) and a generated declaration $\hat{d}$, they are said to be matched if $d = \hat{d}$. Let $\mathcal{D}$ and $\widehat{\mathcal{D}}$ denote the sets of reference and generated declarations, respectively. A False Positive (FP) is a generated declaration $\hat{d}$ that is unmatched, while a False Negative (FN) is a reference declaration $d$ that is unmatched. The canonical accuracy is defined as:

$$1 - \frac{\min(|\text{FP}| + |\text{FN}|, |\mathcal{D}|)}{|\mathcal{D}|}$$

where any score under 100% indicates that the formulations are not equivalent.

Canonical accuracy imposes a strong assumption that generated MILPs must adhere to the same variable order as the ground-truth MILP. As illustrated in Figure 1, if the constraints in $\alpha$ are permuted differently from those in $\alpha'$, they are erroneously treated as nonequivalent, despite being functionally identical. More broadly, canonical accuracy fails in cases where the two formulations differ based on variable or constraint permutations.

**Execution accuracy** captures whether two optimization problems have the same optimal objective value (AhmadiTeshnizi et al., 2024).

**Definition 3.2** (Execution Accuracy). $\alpha$ and $\alpha'$ are considered equivalent if $z^* = z'^*$.

Execution accuracy has a clear limitation: it is not robust to rescaling, a common transformation in MILPs that may simply reflect a change in units. For example, in Figure 1, the objective function in $\alpha'$ is rescaled, which would lead execution accuracy to incorrectly classify $\alpha$ and $\alpha'$ as nonequivalent.

**Example:** *Given a graph $G = (V, E)$, where $V$ represents the set of vertices and $E$ represents the set of edges, find the largest stable set, where a stable set is a subset $S \subseteq V$ such that no two vertices in $S$ are adjacent (i.e., there is no edge $(u, v) \in E$ with $u, v \in S$).*

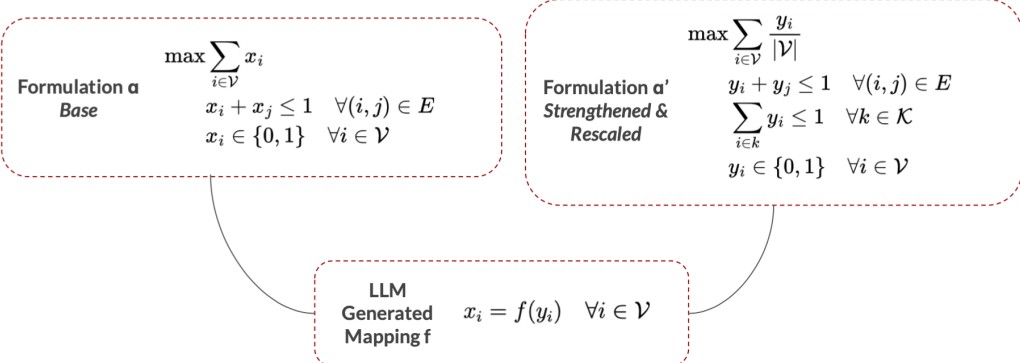

Figure 1: A classic stable set problem, where the two formulations correspond to the same problem description. Formulation $\alpha$ uses the standard formulation, while formulation $\alpha'$ rescales the objective function and adds cutting planes based on cliques (where $\mathcal{K}$ denotes the set of cliques in $G$). LLMs are used to find the mapping function $f$ that maps the variables of $\alpha'$ into the variable space of $\alpha$. An example mapping would be the identity function $f(y_i) = y_i$.

Previous studies have shown that MILPs can be represented as bipartite graphs (Chen et al., 2023a;b; Khalil et al., 2017; Gasse et al., 2019), providing a foundation for defining equivalence using graph-isomorphism based approaches such as the **WL-test**. To construct this bipartite graph, a node is added for each variable and each constraint of the graph. An edge connects a variable node to a constraint node if that variable has a non-zero coefficient in the corresponding constraint. The nodes and edges are endowed with various real-valued attributes describing the MILP (for example, a variable node's attributes will include its coefficient in the objective function). The WL-test tests whether two graphs are isomorphic.

**Definition 3.3** (WL-test (Douglas, 2011)). Let $G = (V, E)$ and $H = (V', E')$ be two graphs. The Weisfeiler-Lehman test is an iterative label refinement procedure used to determine whether $G$ and $H$ are distinguishable. Initially, each vertex $v \in V$ is assigned a label $\ell_0(v)$. At each iteration $t$, the label of each vertex $v$ is updated as follows:

$$\ell_{t+1}(v) = \text{hash}\left(\ell_t(v), \{\ell_t(u) \mid u \in \mathcal{N}(v)\}\right)$$

where $\mathcal{N}(v)$ denotes the set of neighbors of $v$, and the function hash$(\cdot)$ provides a unique encoding neighboring nodes' labels. The process continues iteratively until convergence. To compare graphs, the WL-test computes the multisets of final labels for $G$ and $H$. If these multisets differ at any iteration, the graphs are determined to be non-isomorphic, which indicates that they are not equivalent.

Modifications of the WL-test were proposed by Wang et al. (2024) to evaluate formulation equivalence. Xing et al.

(2024) also introduced a related method based on graph-edit distance, which is a softer version of the WL-test. Since graph-based methods evaluate equivalence after transforming formulations $\alpha$ and $\alpha'$ into bipartite graphs, they will treat the two formulations as non-equivalent if structural modifications change the number of variables or constraints. Such modifications are extremely common (and desired) in MILPs, as techniques like adding cutting planes, reformulating constraints, or introducing auxiliary variables are frequently used to improve solver efficiency and tighten linear relaxations. For example, the second formulation in Figure 1 includes *clique cutting planes*:

$$\sum_{i \in k} y_i \leq 1 \quad \forall k \in \mathcal{K}$$

for cliques $k \in \mathcal{K}$ in the graph. These cutting planes are well-known to strengthen the linear relaxation of the stable set MILP formulation (Conforti et al., 2014).

### 3.2. MILP Equivalence Based on Karp Reduction

Towards a more formal notion of MILP formulation equivalence, we introduce a new definition inspired by a classical tool from complexity theory called a *Karp Reduction*:

**Definition 3.4** (Karp Reduction). Two decision problems $\mathcal{P}, \mathcal{Q}$ are said to be equivalent if there exists a function $f$ that maps *arbitrary instances* of $\mathcal{P}$ to $\mathcal{Q}$ such that:

- If $p$ is a yes-instance of $\mathcal{P}$, then $f(p)$ is a yes-instance of $\mathcal{Q}$,

- If $p$ is a no-instance of $\mathcal{P}$, then $f(p)$ is a no-instance of $\mathcal{Q}$, and

- $f$ can be computed in polynomial time.

A Karp reduction can be used to show that two decision problems are equivalent (i.e., a solution to one can be used to find a solution to the other). These reductions hold for arbitrary instances of the two decision problems, but we leverage a similar approach to establish the equivalence between two specific formulations of an MILP problem instance. Consider two optimization problem formulations $\alpha$, $\alpha'$ that correspond to the same optimization problem $\mathcal{P}$. Our goal is to formally check that an optimal solution to one formulation can be used to generate an optimal solution to the other formulation for a specific instantiation of the problem. Unlike traditional Karp reductions, which define mappings for arbitrary instances, we focus on *instance-specific* mappings. Moreover, our approach maps between solutions of the optimization problem rather than the instance itself.

We also relax the condition that a no-instance (which corresponds to an infeasible or suboptimal solution) under one formulation needs to be mapped to a no-instance of the other. This distinction is important in settings where a MILP formulation may exclude some, but not all, optimal solutions to improve efficiency. For example, adding symmetry-breaking constraints to an optimization model is a common modeling practice that removes functionally equivalent solutions. With these distinctions in mind, we formalize a new notion of equivalence for MILP formulations which we call *Quasi-Karp Equivalence*:

**Definition 3.5** (Quasi-Karp Equivalence). Suppose $\alpha$ and $\alpha'$ are two optimization problems over $\mathbb{R}^d$ and $\mathbb{R}^{d'}$, respectively. We say $\alpha'$ is *Quasi-Karp equivalent* to $\alpha$ if there exists an algorithm $\mathcal{A}(\alpha, \alpha')$ that produces a mapping $f : \mathbb{R}^{d'} \to \mathbb{R}^d$ such that:

- If $x^*$ is an optimal solution to $\alpha'$, then $f(x^*)$ is an optimal solution to $\alpha$,

- $f$ can be computed in polynomial time, and

- $\mathcal{A}(\alpha, \alpha')$ runs in polynomial time for all $\alpha, \alpha'$.

Note that the defintion of Quasi-Karp equivalence is *directional*, meaning that $\alpha'$ being Quasi-Karp equivalent to $\alpha$ does not necessarily imply that $\alpha$ is Quasi-Karp equivalent to $\alpha'$. Also note there is a distinction between the second and third point in definition 3.5: it is possible for $\mathcal{A}$ to run in polynomial time (e.g., a program implementing $f$), but for $f$ itself to require super-polynomial time to evaluate. For example, $\mathcal{A}$ could construct a branch-and-bound solver as $f$-in which case $\mathcal{A}$ runs in polynomial time, but $f$ may not.

In Figure 1, an example of one such mapping $f$ would be $x_i = y_i, \forall i \in \mathcal{V}$, which is a linear function. Intuitively, the notion of *Quasi-Karp Equivalence* is meaningful only when the optimization problem is NP-hard and both optimization formulations admit feasible solutions with finite

---

**Algorithm 1** *EquivaMap*

1: **Input:** Two optimization formulations $\alpha, \alpha'$ with objective $\min c^\top x$, $\min c'^\top x'$ respectively. A solver $S$ that finds an optimal solution $x^*$ for $\alpha$ and $x'^*$ for $\alpha'$.
2: **Output:** A Boolean value indicating whether $\alpha$ and $\alpha'$ are Quasi-Karp equivalent.
3: # {Call an LLM with instance-dependent prompt to find a mapping}
4: $f \leftarrow \mathcal{A}(\alpha, \alpha')$
5: # {Obtain an optimal solution of $\alpha'$ using solver $S$}
6: $x'^* \leftarrow S(\alpha')$
7: # {Map the solution $x'^*$ to a candidate solution in $\alpha$}
8: $\hat{x} \leftarrow f(x'^*)$
9: # {Check if $\hat{x}$ is optimal and feasible for $\alpha$}
10: **if** $c^\top \hat{x} = c^\top x^*$ and $\hat{x}$ feasible for $\alpha$ **then**
11:      **return** True
12: **else**
13:      **return** False
14: **end if**

---

optimal values. If both formulations are infeasible, then neither has a valid solution, making any comparison between them trivial and uninformative. Declaring two infeasible problems equivalent does not provide any insight into their structural or computational properties. Likewise, if one formulation is infeasible while the other is feasible, then no valid mapping $f$ can transform an optimal solution of one into the other. Finally, if a formulation is unbounded, then it lacks a finite optimal solution, so no single "optimal" point can be mapped from one formulation to another. Thus, we use *Quasi-Karp Equivalence* to check equivalence between feasible, bounded formulations.

### 3.3. EquivaMap: LLM-Based Mapping Discovery with Lightweight Verification

To determine the mapping between $\alpha$ and $\alpha'$, we propose *EquivaMap*, a framework that leverages LLMs as the map-finding algorithm $\mathcal{A}$ from Definition 3.5. Specifically, given two formulations $\alpha$ and $\alpha'$ corresponding to a given instance of the problem $\mathcal{P}$, the algorithm $\mathcal{A}$ returns a mapping function $f$ that aligns their solutions: $f = \mathcal{A}(\alpha, \alpha')$.

In *EquivaMap* (Algorithm 1), we first use the LLM $\mathcal{A}$ to find the mapping $f$ for the pair of formulations $(\alpha, \alpha')$ using an instance-specific prompt (Appendix A). Using a solver $S$, we compute an optimal solution $x'^*$ to $\alpha'$. With $f$, we obtain a candidate solution $\hat{x} = f(x^*)$ for $\alpha$. We verify whether $\hat{x}$ is an optimal solution of $\alpha$ by substituting $\hat{x}$ into $\alpha$ and verifying that $\hat{x}$ is feasible and optimal (i.e., $c^\top \hat{x} = c^\top x^*$).

A key component of *EquivaMap* is the instance-specific prompt, which guides the LLM in finding the mapping function $f$. The prompt includes a structured description of

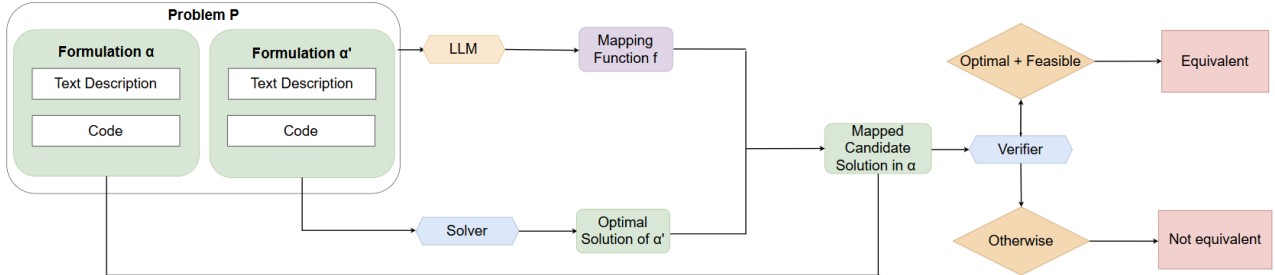

Figure 2: Workflow of *EquivaMap*. The method evaluates equivalence between two formulations ($\alpha$ and $\alpha'$) of the same optimization problem instance $\mathcal{P}$. An LLM generates a mapping function ($f$) to map between the variable spaces of $\alpha$ and $\alpha'$. The mappings are applied to transform the optimal solution of $\alpha'$ into a candidate solution in $\alpha$. A verifier assesses whether the candidate solution is feasible and optimal for $\alpha$. If the verification succeeds, $\alpha$ and $\alpha'$ are deemed equivalent; otherwise, they are classified as not equivalent.

each variable in the formulation ($\alpha$), including its textual description, the constraints in which it appears, and whether it appears in the objective function. Note that if a variable is defined over a set (e.g., $x_i \ \forall i \in \mathcal{V}$ in Figure 1), the definition of the variable is only included once in the prompt. This allows the prompt to scale with the number of *sets of variables*, which can be far less than the number of individual variables for large-scale problems (see Appendix C for an example). We provide an analogous description of the formulation ($\alpha'$). The prompt then instructs the LLM to generate a linear mapping for each variable in $\alpha$, expressed as a list of coefficients and corresponding variable names in $\alpha'$. For more details, we defer to Appendix A.

Below we illustrate this procedure using the example in Figure 1. Suppose the optimal solution of $\alpha'$ is $y_i^*$ for all $i \in \mathcal{V}$. Applying the identity mapping function $f$, we compute $\hat{x}_i = f(y_i^*) = y_i^*$ for all $i \in \mathcal{V}$. We confirm $\hat{x}$ is feasible, and substitute $\hat{x}_i$ into the objective function

$$\sum_{i \in \mathcal{V}} x_i,$$

to verify whether

$$\sum_{i \in \mathcal{V}} \hat{x}_i = \sum_{i \in \mathcal{V}} x_i^*.$$

Note that the mapping $f$ discovered by *EquivaMap* is not instance-specific but operates at the formulation level. We feed the LLM symbolic formulations where parameters and sets, such as the graph $G = (V, E)$, remain abstract instead of being replaced by real values. The LLM then infers a symbolic mapping between the two formulations that is applicable across all instances. A more explicit prompt example can be found in Appendix B. However, the verification step (L9-14) in our algorithm is instance-specific: we instantiate the symbolic formulation with concrete parameter values and sets and verify that the mapped solution

is valid. Thus, while the mapping is over formulations, the verification check is over instances.

Comparing *EquivaMap* to Definition 3.5 of Quasi-Karp Equivalence, we note that, under the reasonable assumption that the LLM's inference time is polynomial in the length of its input prompt, *EquivaMap* runs in polynomial time. Moreover, by restricting the mapping function $f$ to be linear, we ensure that it can be computed in polynomial time.

**Stochasticity in LLMs and Aggregation.** Since LLMs have stochastic outputs, we run Algorithm 1 $K$ times and then aggregate the outputs by declaring ($\alpha, \alpha'$) equivalent if at least one of the $K$ attempts produces a valid mapping.

## 4. Experiments

We conduct a comprehensive evaluation of *EquivaMap* by introducing *EquivaFormulation* — to the best of our knowledge, the first dataset that contains equivalent formulations of MILP instances. Moreover, the dataset includes details about the transformations used to create these equivalent formulations (Section 4.1).

Next, we evaluate *EquivaMap* on this dataset and compare its performance against established baselines including canonical accuracy, execution accuracy, and the WL-test (Section 4.2).

### 4.1. EquivaFormulation: a dataset of equivalent MILP formulations

We construct *EquivaFormulation* based on the NLP4LP dataset (AhmadiTeshnizi et al., 2024). NLP4LP comprises a diverse set of optimization problems with distinct problem sizes, objective functions, and constraints. Each instance in NLP4LP is composed of three components: (1) A description file with a high-level description of the problem in natural language. (2) An information file which con-

Table 1: Overview of the equivalent and nonequivalent transformations between formulations considered in *EquivaFormulation*. **Transformation Name** describes the type of transformation; **How It Is Transformed** explains the modification applied to the problem; **Example (Before/After)** provides a short snippet demonstrating the difference; **Equivalent?** indicates whether the transformation preserves the original problem's optimal solutions; and **Size** shows the number of affected instances, reported as the count of LP and MILP problems.

| Transformation Name | How It Is Transformed | Example (Before/After) | Equivalent? | Size |
|---|---|---|---|---|
| **Substitute Objective Functions** | Replace objective function $\min c^\top x$ with an auxiliary variable $z$, adding new constraint $z = c^\top x$ | **Before:** $\min c^\top x$ 
 **After:** $\min z$, s.t. $z = c^\top x$ | Yes | 92LP + 140MILP |
| **Add Slack Variables** | Transform constraint $g(\mathbf{x}) \leq b$ into $g(\mathbf{x}) + s = b,\ s \geq 0$ | **Before:** $x + 2y \leq 5$ 
 **After:** $x + 2y + s = 5,\ s \geq 0$ | Yes | 59LP + 134MILP |
| **Replace by Base-10 Representation** | Express an integer variable $N$ in its decimal expansion | **Before:** $x \leq 10^6$ 
 **After:** $x = \sum_{i=0}^{6} d_i \cdot 10^i,\ 0 \leq d_i \leq 9,\ d_i \in \mathbb{Z}$ | Yes | 44LP + 123MILP |
| **Add Valid Inequalities** | Include cutting planes or valid linear combinations that do not exclude any integer feasible solution | **Before:** $\{\, x + 2y \leq 3,\ x \leq 1.5 \,\}$ 
 **After:** $\{\, x + 2y \leq 3,\ x \leq 1.5,\ 2x + 2y \leq 4.5 \,\}$ | Yes | 92LP + 142MILP |
| **Rescaling** | Change units/scales for variables or objectives (e.g., hours to minutes) | **Before:** $x$ (hours) 
 **After:** $60x'$ (minutes) | Yes | 60LP + 133MILP |
| **Replace by Linear Combinations** | Decompose a variable $x$ into $x = x^+ - x^-$ with $x^+, x^- \geq 0$ | **Before:** $x$ 
 **After:** $x^+ - x^-$ | Yes | 77LP + 115MILP |
| **Random Order** | Substitute the original instance with a completely unrelated, randomly chosen instance | **Before:** $\min z$, s.t. $z = c^\top x$ 
 **After:** $\max y$, s.t. $y = 3$ | No | 87LP + 142MILP |
| **Loose Constraints** | Delete certain constraints that are tight at the optimum, altering the feasible set | **Before:** $x + 2y \leq 3$ (binding) 
 **After:** remove $x + 2y \leq 3$ | No | 53LP + 120MILP |
| **Feasibility** | Turn both the original and a randomly chosen instance into feasibility problems (replace objectives with 0) | **Before:** $\min 0$, s.t. $z = c^\top x$ 
 **After:** $\max 0$, s.t. $y = 3$ | No | 87LP + 142MILP |

tains the corresponding mathematical formulation of the optimization instance, written in LaTeX. (3) A file that contains the GurobiPy code corresponding to the mathematical formulation.

As discussed in Section 3.2, we carefully select optimization problems in the NLP4LP dataset by removing the infeasible and unbounded instances. In *EquivaFormulation*, we introduce seven (Quasi-Karp) equivalent transformations and three non-equivalent transformations to transform the formulations in NLP4LP to corresponding (Quasi-Karp) equivalent and nonequivalent counterparts, respectively (Table 1). Our proposed equivalent transformations capture widely used and important modeling techniques in MILPs. Standard practices such as substituting the objective function, adding slack variables, and decomposing variables into positive and negative components help simplify constraints and enforce non-negativity. Additionally, robustness to rescaling is crucial, as quantities can be represented in different units (e.g., 1 $kg$ rather than 1000 $g$). Similarly, certain structural transformations — such as adding valid inequalities, reformulating constraints, or introducing auxil-

iary variables — are commonly employed to enhance solver efficiency and tighten relaxations while preserving the formulation's optimal solution. Finally, we incorporate non-equivalent transformations to evaluate the susceptibility of equivalence-checking methods to false positives. These selected transformations in *EquivaMap* are designed to test the robustness of different equivalence-checking methods in handling diverse MILP formulations.

Since the selected transformations in *EquivaFormulation* are deterministic, to prevent the mapping finder or other equivalence-matching methods from exploiting shortcuts—such as mapping decision variables based on their order (e.g., if variable names are assigned alphabetically)—we apply several transformations to the formulation before processing. Specifically, we randomly permute the order of problem parameters, decision variables, and constraints in the information file. Additionally, we assign distinct names to all decision variables and use GPT-4o to generate varied natural language descriptions for them. These transformations are implemented to reduce the similarity between formulation $\alpha$ and $\alpha'$, ensuring that LLMs cannot exploit

**Table 2:** Accuracy of equivalence-checking methods on formulations obtained from equivalent and non-equivalent transformations.

| | Canonical Acc. | Execution Acc. | WL-test | naive-LLM | EquivaMap |
|---|---|---|---|---|---|
| **Equivalent Transformations** | | | | | |
| **Worst Case** | 0% | 0% | 0% | 3.3% | **100%** |
| Substitute Objective Functions | 0% | **100%** | 0% | 91.2% | **100%** |
| Add Slack Variables | 0% | **100%** | 0% | 36.1% | **100%** |
| Replace by Base-10 Representation | 0% | **100%** | 0% | 53.1% | **100%** |
| Add Valid Inequalities | 0% | **100%** | 0% | 3.3% | **100%** |
| Rescaling | 0% | 0% | 0% | 69.9% | **100%** |
| Replace by Linear Combinations | 0% | **100%** | 0% | 24.4% | **100%** |
| **Non-Equivalent Transformations** | | | | | |
| **Worst Case** | **100%** | 0% | **100%** | 93.6% | **100%** |
| Random Order | **100%** | **100%** | **100%** | 98.7% | **100%** |
| Loose Constraints | **100%** | **100%** | **100%** | 93.6% | **100%** |
| Feasibility | **100%** | 0% | **100%** | **100%** | **100%** |

recognizable transformation patterns to deduce the mapping directly.

## 4.2. Performance

We use GPT-4 (Achiam et al., 2023) as the mapping finder in *EquivaMap*, and evaluate our method against existing baselines, plus a naive LLM baseline (naive-LLM). The naive-LLM baseline uses a prompt that includes two formulations $\alpha$ and $\alpha'$ and directly checks if they are equivalent. The prompt can be found in Appendix A. We set $K = 3$, and report the accuracy as the percentage of paired formulations $\alpha$ and $\alpha'$ that are correctly identified as equivalent or nonequivalent, and summarize the results in Table 2.

The results demonstrate that our method consistently outperforms all baseline approaches, achieving perfect or near-perfect accuracy in almost every scenario. Notably, *EquivaMap* performs perfectly even in cases where all baseline approaches completely fail.

For the equivalent transformations (Table 3), our method performs exceptionally well under challenging transformations, such as *Add Valid Inequalities*, *Rescaling*, and *Replace by Linear Combinations*. Execution accuracy and the WL-test fail universally in these settings, achieving 0% accuracy across all variations. In contrast, *EquivaMap* achieves 100% accuracy. These transformations are critical test cases because they highlight the fundamental weaknesses of existing approaches: they struggle with capturing key modeling techniques such as cutting planes and variable rescaling. For example, execution accuracy fails at *Rescaling*, since a naive solver run does not recognize scaled problem instances as equivalent, while WL-test fails at *Add Valid Inequalities* or *Replace by Linear Combinations* since these transformations break isomorphisms in the graph representation. Our method reliably handles them, achieving near-perfect

accuracy.

The results for non-equivalent variations further highlight the reliability of our method. For the *Feasibility* transformation, execution accuracy fails with 0% accuracy, yet our method achieves perfect accuracy for all instances. Under other non-equivalent settings, our method also works well, achieving 100% accuracy.

We additionally perform a runtime analysis comparing *EquivaMap* with baseline methods such as Execution Accuracy and the WL-test. The results, which show that EquivaMap operates within seconds, are presented in Appendix D.

**Key observations.** *EquivaMap* outperforms all existing equivalence-checking methods, including the naive-LLM baseline. This highlights the necessity of our algorithm — without the explicit map finding and optimality verification steps, naively using LLMs with strong reasoning capabilities will not ensure reliability in checking formulation equivalence. Moreover, these results demonstrate that our method is *reliable* across diverse transformations. It consistently outperforms all baselines, especially in scenarios where execution accuracy and other methods fail.

## 5. Discussion

How can we define the equivalence of two formulations of the same optimization problem instance? In this paper, we address this conceptual gap by proposing Quasi-Karp equivalence and a framework, *EquivaMap*, to systematically check such equivalence. Through extensive experiments on MILP problems, we demonstrate that *EquivaMap* outperforms existing approaches by large. Additionally, by introducing the first well-documented pool of equivalent

optimization formulations, encompassing diverse transformations such as the addition of cutting planes, we provide a valuable dataset for advancing research in this domain.

Beyond examining whether two formulations represent the same problem instance, a promising direction for future research is to leverage *EquivaMap* to check equivalences across diverse optimization problems, such as the equivalence between max-flow and min-cut problems. It is also worth noting that our current work focuses on relatively straightforward transformations, thereby omitting more intricate reformulations such as those emerging from decomposition algorithms. As more complex optimization copilots are developed, we may need equivalence-checking tools that can tackle these more complex transformations.

## Impact Statement

Our framework for checking the formulation equivalence of combinatorial optimization problems offers the potential to streamline the development and deployment of optimization models across diverse fields, including operations research, logistics, and engineering. By systematically identifying equivalences among different formulations, our approach can reduce redundant effort, promote reproducibility, and accelerate innovation. In particular, the ability to recognize and compare formulations at a finer granularity could serve as a critical building block for "optimization copilots" that assist researchers and practitioners in designing, debugging, and refining complex models. At the same time, these advantages come with important considerations. When computational tools can automatically detect formulation similarities, there is a possibility of overlooking nuanced domain-specific constraints or ethical requirements if they are not explicitly accounted for. Over-reliance on such tools could inadvertently propagate modeling oversights or marginalize expert judgment. Furthermore, democratizing advanced optimization capabilities may amplify existing disparities if access to these methods remains unevenly distributed.

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

# A. Additional Experimental Details and Results

In this section, we segment our main results by problem class (i.e., LP vs. MILP), and equivalence (i.e., equivalent vs. nonequivalent). We also include the fraction of each instance solved correctly for each problem type. Our results are consistent across both LP and MILP instances, highlighting that *EquivaMap* outperforms all baseline methods in all settings.

**Table 3:** Accuracy of equivalence-checking methods on formulations obtained from *equivalent* transformations. Rows are partitioned by whether the problems are linear programming problems (LP) or mixed-integer linear programming problems (MILP). Numbers in parentheses correspond to the raw fraction of instances solved correctly.

| Transformation | Canonical Acc. | Execution Acc. | WL-test | naive-LLM | EquivaMap |
|---|---|---|---|---|---|
| **LP** | | | | | |
| **Substitute Objective Functions** | 0%(0/92) | 100%(92/92) | 0%(0/92) | 94.6%(87/92) | 100%(92/92) |
| **Add Slack Variables** | 0%(0/59) | 100%(59/59) | 0%(0/59) | 49.1%(29/59) | 100%(59/59) |
| **Replace by Base-10 Representation** | 0%(0/44) | 100%(44/44) | 0%(0/44) | 50%(22/44) | 100%(44/44) |
| **Add Valid Inequalities** | 0%(0/92) | 100%(92/92) | 0%(0/92) | 6.5%(6/92) | 100%(92/92) |
| **Rescaling** | 0%(0/60) | 0%(0/60) | 0%(0/60) | 76.7%(46/60) | 100%(60/60) |
| **Replace by Linear Combinations** | 0%(0/77) | 100%(77/77) | 0%(0/77) | 13.0%(10/77) | 100%(77/77) |
| **MILP** | | | | | |
| **Substitute Objective Functions** | 0%(0/140) | 100%(140/140) | 0%(0/140) | 87.9%(123/140) | 100%(140/140) |
| **Add Slack Variables** | 0%(0/134) | 100%(134/134) | 0%(0/134) | 23.1%(31/134) | 100%(134/134) |
| **Replace by Base-10 Representation** | 0%(0/123) | 100%(123/123) | 0%(0/123) | 56.1%(69/123) | 100%(123/123) |
| **Add Valid Inequalities** | 0%(0/142) | 100%(142/142) | 0%(0/142) | 0%(0/142) | 100%(142/142) |
| **Rescaling** | 0%(0/133) | 0%(0/133) | 0%(0/133) | 63.2%(84/133) | 100%(133/133) |
| **Replace by Linear Combinations** | 0%(0/115) | 100%(115/115) | 0%(0/115) | 35.7%(41/115) | 100%(115/115) |

**Table 4:** Accuracy of equivalence-checking methods on formulations obtained from *nonequivalent* transformations. Rows are partitioned by whether the problems are linear programming problems (LP) or mixed-integer linear programming problems (MILP). Numbers in parentheses correspond to the raw fraction of instances solved correctly.

| Transformation | Canonical Acc. | Execution Acc. | WL-test | naive-LLM | EquivaMap |
|---|---|---|---|---|---|
| **LP** | | | | | |
| **Random Order** | 100%(87/87) | 100%(87/87) | 100%(87/87) | 98.9%(86/87) | 100%(87/87) |
| **Loose Constraints** | 100%(53/53) | 100%(53/53) | 100%(53/53) | 88.7%(47/53) | 100%(53/53) |
| **Feasibility** | 100%(87/87) | 0%(0/87) | 100%(87/87) | 100%(87/87) | 100%(87/87) |
| **MILP** | | | | | |
| **Random Order** | 100%(142/142) | 100%(142/142) | 100%(142/142) | 98.6%(140/142) | 100%(142/142) |
| **Loose Constraints** | 100%(120/120) | 100%(120/120) | 100%(120/120) | 96.7%(116/120) | 100%(120/120) |
| **Feasibility** | 100%(142/142) | 0%(0/142) | 100%(142/142) | 100%(142/142) | 100%(142/142) |

# B. Prompts

## B.1. naive-LLM Prompt

Listing 1: naive-LLM Prompt

```
You are given two optimization problem formulations (both declared as MIP).
Decide if they are equivalent formulations.

First problem formulation (Problem A):
{
  "parametrized_description": "A laundromat can buy two types of washing machines, a top-
      loading model and a front-loading model. The top-loading model can wash
      WashRateTopLoading items per day while the front-loading model can wash
      WashRateFrontLoading items per day. The top-loading model consumes
      EnergyConsumptionTopLoading kWh per day while the front-loading model consumes
```

```
      EnergyConsumptionFrontLoading kWh per day. The laundromat must be able to wash at
      least MinItemsPerDay items per day and has available MaxEnergyPerDay kWh per day.
      Since the top-loading machines are harder to use, at most MaxFractionTopLoading of
      the machines can be top-loading. Further, at least MinNumFrontLoading machines
      should be front-loading. How many of each machine should the laundromat buy to
      minimize the total number of washing machines?",
  "keywords": [
    "N.A."
  ],
  "parameters": {
    "WashRateTopLoading": {
      "description": "Number of items washed per day by a top-loading machine",
      "shape": []
    },
    "WashRateFrontLoading": {
      "description": "Number of items washed per day by a front-loading machine",
      "shape": []
    },
    "EnergyConsumptionTopLoading": {
      "description": "Energy consumed per day by a top-loading machine (kWh)",
      "shape": []
    },
    "EnergyConsumptionFrontLoading": {
      "description": "Energy consumed per day by a front-loading machine (kWh)",
      "shape": []
    },
    "MinItemsPerDay": {
      "description": "Minimum number of items to wash per day",
      "shape": []
    },
    "MaxEnergyPerDay": {
      "description": "Maximum available energy per day (kWh)",
      "shape": []
    },
    "MaxFractionTopLoading": {
      "description": "Maximum fraction of machines that can be top-loading",
      "shape": []
    },
    "MinNumFrontLoading": {
      "description": "Minimum number of front-loading machines",
      "shape": []
    }
  },
  "variables": {
    "NumTopLoading": {
      "description": "The number of top-loading machines",
      "type": "continuous",
      "shape": []
    },
    "NumFrontLoading": {
      "description": "The number of front-loading machines",
      "type": "continuous",
      "shape": []
    }
  },
  "constraints": [
    {
      "description": "A top-loading machine washes WashRateTopLoading items per day and a
          front-loading machine washes WashRateFrontLoading items per day. The total
          number of items washed per day must be at least MinItemsPerDay.",
      "formulation": "WashRateTopLoading \\cdot NumTopLoading + WashRateFrontLoading \\
          cdot NumFrontLoading \\geq MinItemsPerDay",
      "code": {
        "gurobipy": "model.addConstr(WashRateTopLoading * NumTopLoading +
            WashRateFrontLoading * NumFrontLoading >= MinItemsPerDay)"
```

```
      }
    },
    {
      "description": "A top-loading machine consumes EnergyConsumptionTopLoading kWh per
          day and a front-loading machine consumes EnergyConsumptionFrontLoading kWh per
          day. The total energy consumption per day cannot exceed MaxEnergyPerDay kWh.",
      "formulation": "NumTopLoading \\times EnergyConsumptionTopLoading + NumFrontLoading
          \\times EnergyConsumptionFrontLoading \\leq MaxEnergyPerDay",
      "code": {
        "gurobipy": "model.addConstr(EnergyConsumptionTopLoading * NumTopLoading +
            EnergyConsumptionFrontLoading * NumFrontLoading <= MaxEnergyPerDay)"
      }
    },
    {
      "description": "At most MaxFractionTopLoading fraction of the total machines can be
          top-loading.",
      "formulation": "NumTopLoading \\leq MaxFractionTopLoading \\times (NumTopLoading +
          NumFrontLoading)",
      "code": {
        "gurobipy": "model.addConstr(NumTopLoading <= MaxFractionTopLoading * (
            NumTopLoading + NumFrontLoading))"
      }
    },
    {
      "description": "At least MinNumFrontLoading machines must be front-loading.",
      "formulation": "NumFrontLoading \\geq MinNumFrontLoading",
      "code": {
        "gurobipy": "model.addConstr(NumFrontLoading >= MinNumFrontLoading)"
      }
    }
  ],
  "objective": {
    "description": "Minimize the total number of washing machines purchased.",
    "formulation": "Min \\ NumTopLoading + NumFrontLoading",
    "code": {
      "gurobipy": "model.setObjective(NumTopLoading + NumFrontLoading, GRB.MINIMIZE)"
    }
  }
}

Second problem formulation (Problem B):
{
  "parametrized_description": "A laundromat can buy two types of washing machines, a top-
      loading model and a front-loading model. The top-loading model can wash V items per
      day while the front-loading model can wash T items per day. The top-loading model
      consumes F kWh per day while the front-loading model consumes A kWh per day. The
      laundromat must be able to wash at least J items per day and has available R kWh per
       day. Since the top-loading machines are harder to use, at most S of the machines
      can be top-loading. Further, at least W machines should be front-loading. How many
      of each machine should the laundromat buy to minimize the total number of washing
      machines?",
  "keywords": [
    "N.A."
  ],
  "parameters": {
    "W": {
      "description": "The smallest quantity of front-loading machines.",
      "shape": []
    },
    "A": {
      "description": "Daily electricity usage of a front-loading washer (kWh)",
      "shape": []
    },
    "R": {
```

```json
      "description": "The highest amount of energy that can be obtained in a single day (
          kWh).",
      "shape": []
    },
    "S": {
      "description": "The highest percentage of machines that can have a top-loading
          feature.",
      "shape": []
    },
    "F": {
      "description": "Daily energy usage of a top-loading washing machine in kilowatt-
          hours",
      "shape": []
    },
    "J": {
      "description": "The smallest quantity of items that need to be cleaned on a daily
          basis",
      "shape": []
    },
    "V": {
      "description": "Quantity of objects cleaned daily using a top-loading washer",
      "shape": []
    },
    "T": {
      "description": "The quantity of objects cleaned daily with a front-loading washing
          machine.",
      "shape": []
    }
  },
  "variables": {
    "a": {
      "description": "The quantity of top-loading appliances",
      "type": "continuous",
      "shape": []
    },
    "g": {
      "description": "The quantity of front-loading machines",
      "type": "continuous",
      "shape": []
    }
  },
  "constraints": [
    {
      "description": "A top-loading washer cleans V items daily, while a front-loading
          washer cleans T items daily. The combined total of items cleaned each day should
           not fall below J.",
      "formulation": "J \\leq V \\cdot a + T \\cdot g",
      "code": {
        "gurobipy": "model.addConstr(V * a + T * g >= J)"
      }
    }
  ],
  "objective": {
    "description": "Reduce the overall quantity of washing machines bought.",
    "formulation": "Min \\ g + a",
    "code": {
      "gurobipy": "model.setObjective(a + g, GRB.MINIMIZE)"
    }
  }
}
```

Based on the data, please respond with exactly one of the following:
- "Equivalent" if these two are the same formulation. Be rigorous in your reasoning.
- "Not Equivalent" if they are different. When you are not sure, say "Not Equivalent".

```
Briefly explain the reasoning in 1-2 sentences, then end with the word "Equivalent" or "
    Not Equivalent" on its own line.
```

### B.1.1. *EquivaMap* PROMPT

#### Listing 2: *EquivaMap* Prompt

```
You are an AI language model assisting in mapping variables between two optimization
    problems by analyzing their roles in constraints and the objective function.

**Variable from Problem 1:**
- **Name:** OdorRemovingChemicalUnits
- **Description:** The number of units of odor-removing chemical used per house
- **Constraints involving OdorRemovingChemicalUnits:**
  - Description: The total number of chemical units used per house cannot exceed
      MaxTotalUnits.
    Formulation: CleansingChemicalUnits + OdorRemovingChemicalUnits \leq MaxTotalUnits
  - Description: The number of cleansing chemical units used cannot exceed
      MaxCleansingToOdorRatio times the number of odor-removing chemical units used.
    Formulation: CleansingChemicalUnits \leq MaxCleansingToOdorRatio \cdot
        OdorRemovingChemicalUnits
- **In Objective Function:** Yes

**Variables from Problem 2:**
- **Name:** v_0
  **Description:** Digit 0 of the The quantity of cleaning solution units utilized per
      household
  **Constraints involving v_0:**
    - Description: The quantity of cleansing chemical units applied must not surpass H
        times the quantity of odor-removing chemical units used.
      Formulation: H \cdot (f_0*10^0 + f_1*10^1) \geq (v_0*10^0 + v_1*10^1 + v_2*10^2)
    - Description: The cumulative quantity of chemical components utilized for each
        residence must not surpass T.
      Formulation: T \geq (f_0*10^0 + f_1*10^1) + (v_0*10^0 + v_1*10^1 + v_2*10^2)
    - Description: The company is required to utilize a minimum of G units of the cleaning
         solution per household.
      Formulation: G \leq (v_0*10^0 + v_1*10^1 + v_2*10^2)
  **In Objective Function:** Yes

- **Name:** v_1
  **Description:** Digit 1 of the The quantity of cleaning solution units utilized per
      household
  **Constraints involving v_1:**
    - Description: The quantity of cleansing chemical units applied must not surpass H
        times the quantity of odor-removing chemical units used.
      Formulation: H \cdot (f_0*10^0 + f_1*10^1) \geq (v_0*10^0 + v_1*10^1 + v_2*10^2)
    - Description: The cumulative quantity of chemical components utilized for each
        residence must not surpass T.
      Formulation: T \geq (f_0*10^0 + f_1*10^1) + (v_0*10^0 + v_1*10^1 + v_2*10^2)
    - Description: The company is required to utilize a minimum of G units of the cleaning
         solution per household.
      Formulation: G \leq (v_0*10^0 + v_1*10^1 + v_2*10^2)
  **In Objective Function:** Yes

- **Name:** v_2
  **Description:** Digit 2 of the The quantity of cleaning solution units utilized per
      household
  **Constraints involving v_2:**
    - Description: The quantity of cleansing chemical units applied must not surpass H
        times the quantity of odor-removing chemical units used.
      Formulation: H \cdot (f_0*10^0 + f_1*10^1) \geq (v_0*10^0 + v_1*10^1 + v_2*10^2)
    - Description: The cumulative quantity of chemical components utilized for each
        residence must not surpass T.
      Formulation: T \geq (f_0*10^0 + f_1*10^1) + (v_0*10^0 + v_1*10^1 + v_2*10^2)
```

   – Description: The company is required to utilize a minimum of G units of the cleaning
      solution per household.
     Formulation: $G \leq (v_0*10^0 + v_1*10^1 + v_2*10^2)$
  **In Objective Function:** Yes

- **Name:** f_0
  **Description:** Digit 0 of the The quantity of odor-neutralizing chemical applied in
    each household
  **Constraints involving f_0:**
   – Description: The quantity of cleansing chemical units applied must not surpass H
     times the quantity of odor-removing chemical units used.
    Formulation: $H \cdot (f_0*10^0 + f_1*10^1) \geq (v_0*10^0 + v_1*10^1 + v_2*10^2)$
   – Description: The cumulative quantity of chemical components utilized for each
     residence must not surpass T.
    Formulation: $T \geq (f_0*10^0 + f_1*10^1) + (v_0*10^0 + v_1*10^1 + v_2*10^2)$
  **In Objective Function:** Yes

- **Name:** f_1
  **Description:** Digit 1 of the The quantity of odor-neutralizing chemical applied in
    each household
  **Constraints involving f_1:**
   – Description: The quantity of cleansing chemical units applied must not surpass H
     times the quantity of odor-removing chemical units used.
    Formulation: $H \cdot (f_0*10^0 + f_1*10^1) \geq (v_0*10^0 + v_1*10^1 + v_2*10^2)$
   – Description: The cumulative quantity of chemical components utilized for each
     residence must not surpass T.
    Formulation: $T \geq (f_0*10^0 + f_1*10^1) + (v_0*10^0 + v_1*10^1 + v_2*10^2)$
  **In Objective Function:** Yes

Based on the above information, find the best mapping from variables in Problem 2 for the
    variable 'OdorRemovingChemicalUnits' from Problem 1. The mapping can be a linear
   combination of variables from Problem 2, possibly with constant multipliers. Your goal
   is to express 'OdorRemovingChemicalUnits' in terms of variables from Problem 2, as
   accurately as possible, based on their roles in the constraints and objective
   functions.

**Important Instructions:**

- **Provide only the mapping for 'OdorRemovingChemicalUnits' as a JSON object.**
- **Do not include any additional text, explanations, or formatting.**
- **The JSON object must follow this exact structure:**

```
{
  "OdorRemovingChemicalUnits": [
    {
      "constant": constant_value_1,
      "variable": "variable_name_1"
    },
    {
      "constant": constant_value_2,
      "variable": "variable_name_2"
    },
    ...
  ]
}
```

- **If there is only one term in the mapping, the list should contain a single object.**
- **Use numerical values for constants (decimals), and enclose variable names in double
   quotes ("").**

**Examples:**

1. If the best mapping is '0.1*a', your response should be:

```
{
  "OdorRemovingChemicalUnits": [
    {
      "constant": 0.1,
      "variable": "a"
    }
  ]
}

2. If the best mapping is '0.1*a + 0.01*b', your response should be:

{
  "OdorRemovingChemicalUnits": [
    {
      "constant": 0.1,
      "variable": "a"
    },
    {
      "constant": 0.01,
      "variable": "b"
    }
  ]
}

3. If the best mapping is a single variable 'a' with a coefficient of 1, your response
   should be:

{
  "OdorRemovingChemicalUnits": [
    {
      "constant": 1,
      "variable": "a"
    }
  ]
}

4. If there is no direct mapping, your response should be:

{
  "OdorRemovingChemicalUnits": [
    {
      "constant": "none",
      "variable": "none"
    }
  ]
}

Please ensure your response is a valid JSON object that can be parsed by standard JSON
    parsers.
```

## C. Maximum Independent Set example

Figure 3: Comparison between set-based and individual-variable representations in JSON input formatting. EquivaMap operates on sets of variables, allowing the metadata and constraints to be described concisely (left) instead of expanding each variable individually, resulting in longer and more redundant prompts (right).

Consider the maximum independent set example in Figure 3. *EquivaMap* takes in set-based representations of input formulations (left). When the prompt iterates between variables of formulations $\alpha$ and $\alpha'$, it processes the entire 'Node' set as input, rather than individual variables like Node 1, Node 2, etc. If variables in formulation $alpha'$ are labeled as Node', the mapping discovered by the LLM will be $Node[i] = Node'[i], \forall i$, instead of separate mappings for each indexed variable. This distinction is crucial for scalability, as it means our prompt size remains constant regardless of the number of nodes in the graph.

## D. Runtime Analysis

To evaluate the computational overhead of *EquivaMap* and the baseline methods, we measured the average runtime across all instances in our dataset. The breakdown of time spent on different components is presented in Table 5.

**Table 5:** Mean ($\pm$ std. dev.) runtime (seconds) per instance for different components of *EquivaMap* and baselines. Runtime is averaged across all instances in the *EquivaFormulation* dataset.

| Method | Solving Time | LLM Call Time | WL-Test Time | Total |
|---|---|---|---|---|
| Execution Accuracy | $0.12 \pm 0.02$ | - | - | $0.12 \pm 0.02$ |
| WL-Test | - | - | $0.38 \pm 0.07$ | $0.38 \pm 0.07$ |
| EquivaMap | $0.12 \pm 0.02$ | $11.88 \pm 4.48$ | - | $12.00 \pm 4.50$ |

While *EquivaMap* exhibits a higher total runtime compared to the baselines, this cost should be considered in light of its significantly improved accuracy and its ability to identify complex mappings that other methods miss (as shown in Section 4.2). The LLM interaction, though currently the bottleneck, enables a level of symbolic reasoning and mapping discovery previously unattainable. It is also worth noting that this runtime is for a single equivalence check. In practice, formulation equivalence checking is often an offline analysis task where a higher runtime can be tolerated in exchange for reliable results. Furthermore, the LLM call time can potentially be reduced with more optimized prompting strategies, the use of smaller fine-tuned models, or by leveraging future advancements in LLM efficiency. Additionally, as discussed in Section 3.3, *EquivaMap*'s prompt length scales with the number of sets of variables rather than individual variables, which helps manage the LLM interaction cost for larger, structured problems.

