# OpenReview forum: "EquivaMap: Leveraging LLMs for Automatic Equivalence Checking of Optimization Formulations"
_ICML.cc/2025/Conference — ICML 2025 poster_

### Official Review · Reviewer_Nbcm · 2025-03-14

**Overall Recommendation:** 3

**Summary:**

This paper addresses the challenge of checking equivalence among modeling formulations. The authors introduce quasi-Karp as an equivalence criterion for determining model equivalence. The primary concept of this criterion involves verifying the existence of "equivalence mapping" (quasi-Karp) between models. The authors leverage GPT-based methods to automatically identify the existence of these equivalencies. The dataset used in experiments is derived from the NL4OPT dataset, further crafted by introducing equivalent and non-equivalent transformations. Experimental results indicate that Equivamap achieves 100% accuracy on this dataset.

**Claims And Evidence:**

Not sufficiently clear and convincing.

The experiments are restricted to a dataset derived from the NL4OPT dataset, crafted by introducing equivalent and non-equivalent transformations. Broader experimentation or testing on diverse datasets would enhance result validity and generalizability.

Further, the experiments only show that

**Essential References Not Discussed:**

Not found yet.

**Experimental Designs Or Analyses:**

See **Claims And Evidence** and **Methods And Evaluation Criteria**.

**Methods And Evaluation Criteria:**

No, not enough.

NL4OPT is a relatively simple dataset and there are more complex, and closer-to-reality ones, like MAMO (ComplexLP) and IndustryOR. The generalizability of EquivaMap to more challenging problems remains uncertain.

**Other Comments Or Suggestions:**

NA

**Other Strengths And Weaknesses:**

Other Strengths:
- This work tries to address an important challenge in modeling equivalence checking, where previous works have not provided sufficiently satisfying solutions.

Other Weakness:
- The method faces scalability issues. As the dimensionality of the problem increases, the input prompt length could significantly grow.
- The method also faces reliability issues. The reliance on LLMs introduces a risk of hallucinations, where the model might generate plausible but incorrect equivalence mappings. Such inaccuracies can be challenging to detect, potentially impacting the reliability of Equivamap in more complex or realistic scenarios.

**Questions For Authors:**

In practice, when using the proposed method to benchmark LLMs for optimization modeling, can the proposed method provide better accuracies than the existing method? Specifically on more complex datasets like MAMO (ComplexLP) and IndustryOR.

**Relation To Broader Scientific Literature:**

The ideas of the equivalent transformations (used in the construction of the dataset) are from the optimization area (MILP).

**Theoretical Claims:**

No proof is involved.

The definition of quasi-Karp equivalence is sound and reasonable. If the mapping defined in quasi-Karp equivalence exists, then the two optimization models are equivalent. This claim is true by definition, without the need for separate proof.

---

> ### Author Rebuttal · Authors · 2025-04-01
>
> We sincerely thank the reviewer for the detailed feedback and for acknowledging the importance of our work:
>
> > "This work tries to address an important challenge in modeling equivalence checking, where previous works have not provided sufficiently satisfying solutions."
>
> and for recognizing the soundness of our definition:
>
> > "The definition of quasi-Karp equivalence is sound and reasonable."
>
> We appreciate the opportunity to clarify aspects that may have been misunderstood. While we recognize the importance of evaluating generalization and scalability, we note that some of the concerns—particularly regarding the **MAMO** and **IndustryOR** datasets—**go beyond what these datasets can currently support**. They **do not offer the mathematical formulations and/or solver-compatible code needed for EquivaMap** to systematically compare different instantiations of the same optimization problem. Below, we address these points and provide additional context.
>
> ---
>
> > **C1**: Generalizability to More Complex Datasets (e.g., MAMO (ComplexLP), IndustryOR)
>
> **A1**: We appreciate the reviewer’s interest in evaluating EquivaMap on more datasets. However, datasets like **MAMO (ComplexLP)** and **IndustryOR** are not compatible with our evaluation method. While these datasets provide natural language problem descriptions and the respective optimal objective values, they do not include the corresponding **mathematical formulations** or **solver code** for obtaining such values. Our framework evaluates equivalence in formulations, thus any viable dataset for assessing it requires the existence of the formulations.
>
> Specifically, EquivaMap tests quasi-Karp equivalence by identifying and verifying mappings between **different mathematical formulations** of the same optimization problem. This process requires access to mathematical formulations (or their equivalent solver-accessible code). Since **MAMO (ComplexLP)** and **IndustryOR** provide neither, they cannot be used to assess our method.
>
> Though the original datasets cannot be used to assess EquivaMap, as an initial step to assess our method’s performance on MAMO and IndustryOR, we **manually constructed the mathematical formulations and solver-compatible code** of five instances from each dataset (we made sure to choose instances with different underlying formulations). We then applied the equivalent and non-equivalent transformations (described in Section 4) and evaluated EquivaMap accordingly. In all cases, EquivaMap achieved **100% accuracy**. We will add this discussion to our paper.
>
> We also respectfully disagree that the **NLP4LP** dataset, which contains **over 300 instances**, is too simplistic for evaluating equivalence mapping. The average **description length** of **NLP4LP (hard)** exceeds 900 characters, indicating rich semantic content. It also includes **multi-dimensional parameters**, like MAMO (ComplexLP) and IndustryOR. This information is shown in Table 1 of [1].
>
> [1] AhmadiTeshnizi, A., et al. (2024). Optimus-0.3: Using LLMs to model and solve optimization problems at scale.
>
> > **C2**: Scalability with the increase of the dimensionality of the problem and corresponding input prompt length
>
> **A2**: A key feature of EquivaMap is that it operates over **sets of variables**, rather than individual variables in a large-scale instance.
>
> For example, in a graph coloring problem, although there is one decision variable per node, the formulation defines a **set of variables** (e.g., x[i] for each node i). EquivaMap only needs to reason about how one such variable set maps to another, rather than processing all individual variables (e.g., x[1], x[2], …) one-by-one. This is reflected in our input JSON format, where each set of variables and constraints is represented once, along with its index set.
>
> This enables scalability because the **number of sets of variables** is typically **much smaller than the total number of variables**. In contrast, approaches that rely on analyzing variable-constraint graphs (e.g., in the WL-test) must operate over formulations that account for the total number of variables, leading to significant overhead as the problem size increases.
>
>  > **C3**: Reliability on LLMs introduces a risk of hallucinations, where the model might generate plausible but incorrect equivalence mappings
>
> **A3**: EquivaMap is designed to be robust to LLM hallucinations. In EquivaMap, for a given LLM-found mapping, we run it through a **verification step** (L239 - L245 right column) that checks whether the mapped solution preserves optimality. Mappings that fail are discarded. This sampling + verification pipeline **ensures correctness** even when the LLM is imperfect.
>
> This distinguishes EquivaMap from **naive prompting-based approaches** that treat LLM output as inherently reliable. Our method instead treats the LLM as a heuristic mapping generator, embedded within a strict verification loop that ensures only correct mappings are accepted.

---

> > ### Comment · Reviewer_Nbcm · 2025-04-05
> >
> > Thank authors for the reply, but I am not fully convinced by the rebuttal.
> >
> > ## Concern 2: Scalability
> >
> > I disagree with the authors' argument, as I believe it may mislead readers:
> >
> > 1. **Prompt length will increase**: As shown in Appendix A and the codes, **the prompt clearly includes every individual variable and constraint**. That means the prompt length must grow significantly as the problem size increases.
> > 2. **EquivaMap needs to examine individual variable/constraints**: According to the paper, EquivaMap should be able to determine whether an added cutting plane is valid. In Figure 1, a cutting plane is added based on a clique $\mathcal{K}$. A clique is a subset of nodes in which every two nodes are adjacent. This means that, to verify whether $\mathcal{K}$ is truly a clique, EquivaMap needs to check $k$ variables and $k(k-1)/2$ constraints. This CANNOT be achieved by simply inspecting the description on the sets, "where each set of variables and constraints is represented once, along with its index set" (mentioned in the rebuttal).
> >
> > ## Concern 1: Dataset
> > I apologize for mistakenly referring to **NLP4OPT** instead of **NLP4LP** in my comment. It was an honest error, and I did not intend to imply that NLP4LP is a simple dataset. However, this brings up two issues:
> > 1. EquivaMap explicitly requires mathematical formulations, which could limit its applicability to datasets where such formulations are not readily available.
> > 2. Although NLP4LP includes challenging instances (e.g., complicated text descriptions), the problem sizes are generally small. Many instances have fewer than 10 variables (based on my initial review of the data). Furthermore, the NLP4LP paper states that "passing all problem information directly to an LLM is not a scalable solution," and they ”separate the numerical data from the problem descriptions, providing only metadata to the LLM“. In contrast, EquvaMap includes all numerical data in the prompt, further enhancing the concern about its scalability.
> >
> > ## Concern 3: Reliability
> >
> > Thanks for the clarification. EquivaMap indeed involves a verification step. Upon re-reading the paper, I realize the concern is more complicated than I initially thought:
> >
> > 1. According to Definition 3.5, Quasi-Karp Equivalence requires that **every** optimal solution of $\alpha'$ can be converted to an optimal solution of $\alpha$. However, **EquivaMap only verifies one optimal solution of $\alpha'$**. It is possible that $\alpha'$ has a strictly larger optimization set, and thus cannot be regarded equivalent to $\alpha$. I did not find explanations about this gap in the original paper.
> >
> > 2. Under the current pipeline, I am unsure whether EquivaMap is truly necessary for the following transformations: Add Slack Variables, Replace by Base-10 Representation, Add Valid Inequalities, Replace by Linear Combinations, Random Order, Loose Constraints. In all these cases, the optimal value will remain unchanged. As shown in the experiments, Execution Acc. already achieves 100% accuracy over all these cases.
> >
> > For instance, consider Add Valid Inequalities. If the added inequality is valid, then $\alpha'$ and $\alpha$ have the same optimal values. If not, they have different ones. EquivaMap does not seem necessary to distinguish between these two cases.
> >
> > If this is the case, the only situations where EquivaMap is effective over solver evaluation are Substitute Objective Functions and Rescaling. These two cases appear much simpler to handle and do not require mixing with the other complicated but solver-differentiable scenarios.
> >
> > 3. BTW, I am not clear about the Feasibility Transformation. The description in Table 1 says this transformation "Turn both the original and a randomly chosen instance into feasibility problems (replace objectives with 0)". But the example is to turn a feasibility problem into another one. Also, I don't understand why solvers cannot differentiate this case.
> >
> > 4. For large-scale and more complex MILP problems, solvers may not always output an optimal solution, or may take an unreasonably long time to obtain results. In such cases, EquivaMap may fail (if the solver fails).
> >
> > ## Concern 4: Data Availability (New)
> > I planned to check the original data to justify Concern 3. However, I cannot find any data in the supplementary material (in the anonymous link). However, the abstract claims: Our **code and data** can be found in this repository..."
> >
> > ## Remark
> > - I notice a significant discrepancy between my score and other reviewers' scores. To facilitate a fair and thorough evaluation of the paper, I re-read the paper, including the Appendix and supplementary materials. After this careful re-examination, I maintain my score..
> > - My research area involves using LLM to address industry-related optimization problems, which often involve hundreds or more variables. As such, I have a particular concern regarding the scalability of methods in this area, which informs my perspective on the paper.

---

> > > ### Author Response · Authors · 2025-04-07
> > >
> > > > C2.1 & 2.2: Scalability
> > >
> > > We think there is still a misunderstanding about our inputs and prompt. Our approach **does indeed operate on sets of variables** (whose size is typically **much smaller than individual variables** for large-scale problems). Consider the max. ind. set example. EquivaMap takes as input the same metadata formatting as NLP4LP:
> > > ```
> > >   "Variables": {
> > >     "Node": {
> > >       "type": "binary",
> > >       "shape": ["NumNodes"]
> > >     }
> > >   },
> > >   "Constraints": {
> > >     "EdgeConstraint": {
> > >       "formulation": "Node[i] + Node[j] <= 1 for all i > j such that Edges[i][j] == 1",
> > >     }
> > >   }
> > > }
> > > ```
> > > In this case, when the prompt (as shown in Appendix A) iterates between variables of $\alpha$ and $\alpha'$, it is taking a set of variables (In this case ```Node```) as input, instead of ```Node[1], Node[2],...```. Suppose in formulation $\alpha'$ the variables are ```Node'```, then the mapping LLM finds will be ```Node[i] = Node'[i] \forall i```, instead of ```Node[1] = Node'[1], Node[2] = Node'[2],...```. Multiple instances in our dataset (No.96, No.132, No.247, etc) also highlight this distinction. This means that our prompt size is the same for this problem whether the underlying graph has 10 nodes or 10000. However, the length of our prompt does increase with the number of **sets of variables**. This is also true for constraints defined over sets (i.e., all cliques). We agree that this distinction was difficult to see in the current draft and we've added more explanation including a full example in a revision.
> > >
> > > We also believe there may be a misunderstanding regarding EquivaMap’s role. EquivaMap is not designed to verify whether a given inequality (e.g., a cutting plane based on a clique) is valid (nor do we claim this in the paper).
> > >
> > > > C1.1: “...could limit its applicability to datasets…"
> > >
> > > EquivaMap is designed specifically for verifying equivalence between two given mathematical formulations. Our focus is not on datasets that lack formulations altogether, but rather on cases where such formulations are available and an equivalence check is needed. We acknowledge that many current benchmarks (e.g., MAMO, IndustryOR) are built around execution accuracy, but this reflects the limitations of available data, not of our method.
> > >
> > > > C1.2: "...the NLP4LP paper states… EquivaMap includes all numerical data in the prompt..."
> > >
> > > We clarify that EquivaMap uses **the same metadata** as NLP4LP. We separate numerical data from textual descriptions and only provide structured metadata to the LLM, as demonstrated in our example addressing C2.1 & 2.2.
> > >
> > > > C3.1: "...EquivaMap only verifies..."
> > >
> > > EquivaMap verifies equivalence using a single optimal solution of the transformed problem. This serves as a **necessary condition** for equivalence under our definition. In practice, this is precisely the case we care about finding the mapping for (i.e., solving $\alpha’$ enables us to solve $\alpha$). We will clarify this in the revision.
> > >
> > > > C3.2: "...Execution Acc. already achieves 100% accuracy..."
> > >
> > > EquivaMap is designed as a **general-purpose evaluator** that does not assume prior knowledge of the transformation type. In real-world scenarios, such transformations are often unknown. Existing approaches can perform well on certain cases; but **without knowing the transformation in advance**, one cannot guarantee their performance. In contrast, EquivaMap achieves 100% accuracy across different transformations **without** relying on such prior knowledge.
> > >
> > > > C3.3: "...Feasibility Transformation..."
> > >
> > > Feasibility transformations replace the objective function of two completely different optimization formulations with **the same constant**, but this does not mean the problems are equivalent—two problems can have the same objective yet entirely different feasible sets. Execution accuracy cannot detect this via optimal values.
> > >
> > > > C3.4: "For large-scale and more complex MILP..."
> > >
> > > We agree that if the solver fails to find any solution, EquivaMap cannot verify equivalence. However, this limitation is shared by execution accuracy, which also relies on solver outputs for generating the optimal objective value. In settings where a solver returns a suboptimal solution, EquivaMap can be used to verify the solution’s feasibility and provide an optimality gap compared to the existing optimal value, which cannot be done by execution accuracy.
> > >
> > > > C4:  "...I cannot find any data..."
> > >
> > > Thank you for pointing this out and we apologize for this! We initially included a link to the Hugging Face repository but later removed it upon realizing it was not anonymous. We have now uploaded the data to the anonymous repository.
> > >
> > > ---
> > >
> > > We emphasize that in response to the reviewer’s earlier suggestions, we ran EquivaMap on instances from MAMO(ComplexLP) and IndustryOR which we manually put into our metadata format and included those results in our rebuttal. We respectfully note that this additional effort was not acknowledged in the reviewer’s most recent comments.

---

### Official Review · Reviewer_u5nU · 2025-03-14

**Overall Recommendation:** 4

**Summary:**

The submission proposed a new method for determining whether two optimization problem formulations are equivalent. The authors introduce Quasi-Karp Equivalence, based on Karp reductions, and propose EquivaMap, a framework that utilizes LLMs to identify mappings between decision variables of different formulations.  The paper also introduces EquivaFormulation, the first open-source dataset of equivalent optimization formulations. Experimental results demonstrate that EquivaMap outperforms existing equivalence-checking methods, including execution accuracy, canonical accuracy, and graph-based approaches. The contributions are positioned within the broader context of optimization copilots and automated mathematical modeling.

**Claims And Evidence:**

The authors provide sufficient evidence to support their claims. However, I have some doubts about the scalability of EquivaMap in practical applications (mathematical modeling copilots), as it requires multiple calls to both the LLM and the solver for each problem instance. Furthermore, we typically seek a formulation that is correct for all problem instances, whereas EquivaMap can only demonstrate instance-specific equivalence, not a general equivalence.

**Essential References Not Discussed:**

The paper cites key relevant works.

**Experimental Designs Or Analyses:**

- The new dataset is well-constructed, covering various equivalence-preserving transformations.
- Baseline comparisons include execution accuracy, canonical accuracy, WL-test, and a naive LLM approach.

**Methods And Evaluation Criteria:**

The methods—embedding an LLM-based “mapping finder” plus a solver-based verification step—are straightforward and well-motivated. However, one drawback of the proposed approach is that it cannot account for partial equivalence between two formulations, which would be beneficial for more refined accuracy metrics. In contrast, methods like canonical accuracy and graph-edit distance do provide a measure of partial similarity, allowing more nuanced evaluations of how closely two formulations align.

**Other Comments Or Suggestions:**

- A breakdown of computational costs (LLM inference time vs. solver time) would provide insights into practical deployment.

**Other Strengths And Weaknesses:**

- The introduction of Quasi-Karp Equivalence is an important conceptual contribution.
- The EquivaFormulation dataset is a valuable resource for future research.
- The work is timely, given the increasing interest in AI-assisted optimization modeling.

**Questions For Authors:**

1 - The identity mapping in Figure 1 appears overly simple and may not effectively illustrate the complexity of the LLM-based mapping step. Could you provide a more challenging example that better demonstrates the depth of reasoning needed to discover a valid mapping?

2 - Because EquivaMap relies on instance-specific mappings, users would have to run the framework for every single problem instance. In addition, handling stochastic LLM outputs requires multiple runs. Do you see this as a serious limitation for real-world usage, especially in time-sensitive scenarios or in modeling copilots?

3 - EquivaMap verifies equivalence only for specific instances, not universally for the entire family of instances. Have you explored or considered approaches that could ensure (or at least give stronger guarantees of) equivalence across all instances rather than one?

4 - The framework currently restricts the mapping function f to be linear. This effectively limits the kinds of transformations EquivaMap can capture. Have you investigated whether more general classes of transformations (e.g., piecewise-linear or polynomial mappings) are feasible, or do you view the linear assumption as a fundamental design choice? If it is fundamental, what types of practically important reformulations do you expect it might fail to handle?

**Relation To Broader Scientific Literature:**

The paper is well-situated within the broader research on LLM-based optimization modeling automation. The discussion on optimization copilots aligns with recent advancements in AI for mathematical modeling.

**Theoretical Claims:**

Quasi-Karp Equivalence: Inspired by polynomial-time Karp reductions, but the formal proof of correctness or completeness is limited to a set of instance-specific transformations. There is no guarantee that every subtlety in modern MILP formulations (such as symmetries or problem-specific decomposition constraints) can be captured by a single linear mapping.

---

> ### Author Rebuttal · Authors · 2025-04-01
>
> We sincerely thank the reviewer for the detailed feedback and for acknowledging the contribution of our work:
>
> > "The introduction of Quasi-Karp Equivalence is an important conceptual contribution."
>
> >” The EquivaFormulation dataset is a valuable resource for future research.”
>
> >” The work is timely, given the increasing interest in AI-assisted optimization modeling.”
>
> ---
>
> Below, we address the questions raised:
>
> > **Q1**: "... Could you provide a more challenging example that better demonstrates the depth of reasoning needed to discover a valid mapping?"
>
> **A1**: Below we present a more challenging transformation as an illustration: a digit-based reformulation where a single bounded integer variable is replaced by its **base-10 representation**. In this case, a bounded integer variable $x \leq 10^6$ is represented by 7 new variables $d_0, d_1, ..., d_6$, each representing one decimal digit of x, along with corresponding bounds and integrality constraints. Thus, the mapping $f$ will map $d_0, d_1, …, d_6$ to $x$, i.e., $x = \sum_{i = 0}^6 10^i d_i$. To successfully identify this mapping, EquivaMap must understand that a single variable $x$ can be **decomposed into multiple variables representing digits**, and recognize the weighted sum structure linking $x$ to its digits. In this case, EquivaMap also achieved **100% accuracy** as demonstrated in our experiments.
>
> > **Q2**: "..., users would have to run the framework for every single problem instance. In addition, handling stochastic LLM outputs requires multiple runs. Do you see this as a serious limitation for real-world usage, especially in time-sensitive scenarios or in modeling copilots?"
>
> **A2**: First, we note that **per-instance evaluation is standard across all existing metrics**, including **canonical accuracy**, **execution accuracy**, and the **WL-test**. This is inherent to the nature of automatic equivalence evaluation of optimization formulations, where ground-truth mappings are known and correctness is assessed instance by instance. Importantly, this is an offline task—it’s not meant to be executed in real-time during modeling. In modeling copilot settings, EquivaMap is meant to be used as an evaluator for a proposed copilot as it can evaluate its performance against correct formulations (instead of being the copilot itself).
>
> We would also like to clarify that compared to execution accuracy, EquivaMap does not require additional MILP solving beyond the original problem instances. In particular, during the final verification step, we only evaluate whether the solution induced by the mapped variables satisfies the constraints in the target formulation and matches the known optimal objective value. This check is computationally lightweight and avoids any further calls to optimization solvers. In other words, once the MILPs are solved initially, **no further solving is needed** to verify a candidate mapping.
>
>
> > **Q3**: "... Have you explored or considered approaches that could ensure (or at least give stronger guarantees of) equivalence across all instances rather than one?"
>
> **A3**: We agree that automatically checking equivalence across an entire family of instances is a powerful and exciting direction! Achieving this would likely require extending our verification process to work on symbolically parameterized formulations, and leveraging formal methods or symbolic solvers to validate that the discovered mapping preserves equivalence under all admissible inputs. We see this as a promising direction for future work.
>
> > **Q4**:  "... Have you investigated whether more general classes of transformations (e.g., piecewise-linear or polynomial mappings) are feasible, or do you view the linear assumption as a fundamental design choice? If it is fundamental, what types of practically important reformulations do you expect it might fail to handle?"
>
> **A4**: Thank you for this thoughtful question! To clarify: EquivaMap does not require $f$ to be linear by design—the only theoretical requirement is that the mapping function $f$ needs to be **polynomial-time computable** (L254 - L264, left column), in line with the notion of quasi-Karp equivalence. In practice, we choose $f$ to be linear because it suffices to cover all the equivalence-preserving transformations we included in our dataset. We will clarify this in our paper.
>
>
> >**C1**: “The methods…  partial equivalence … formulations align.”
>
> **A5**: We appreciate the reviewer raising this point and believe that this could be an interesting future line of work. In many cases, partial equivalence between optimization formulations is not a well-defined or semantically meaningful concept. That is, in these settings, a formulation either **preserves the objective and feasible set**, or it does not, and thus does not have a direct connection to partial equivalence. Precisely identifying when partial equivalence makes sense and how it should be defined is of future interest.

---

### Official Review · Reviewer_FhjQ · 2025-03-17

**Overall Recommendation:** 3

**Summary:**

This paper proposes a new method to assess whether two mathematical MILP formulations of a combinatorial optimization problem are equivalent. To this end, it proposes a formal notion of equivalence inspired by standard Karp reductions, creates a new dataset called EquivaFormulation for assessment, and compares the proposed method against 3 common existing approaches and a naive LLM-based approach. It finds that the proposed method, EquivaMap, works perfectly on the dataset, while all other baselines struggle in some, if not all, categories.

**Claims And Evidence:**

The main claim, that EquivaMap is more effective at recognizing problem formulation equivalence, seems sufficiently supported on the specific classes of transformations considered in EquivaFormulation and in Table 2. It's not clear whether EquivaMap will also work equally well in natural and real settings, such as the motivating one involving that the authors refer to as optimization copilot, or in scenarios where multiple transformations are stacked together. The evaluation does not assess these.

**Essential References Not Discussed:**

N/A

**Experimental Designs Or Analyses:**

Experiments seem justifiable. Results in Table 2 help clearly see which aspects of transformation (different rows) various methods struggle or excel at.

However, as noted above, this evaluation dataset is synthetic and designed specifically around individual transformations. In practice, I suspect one will find combinations of various transformations from one formulation to another. The current evaluation does not test EquivaMap in such compositional transformation scenarios (or on actual instances of transformations seen in the motivating optimization copilot research).

The "worst case" line in Table 2 is a bit confusing and not very helpful. I would suggest dropping it, possibly replacing it with "average" line.

**Methods And Evaluation Criteria:**

Yes

**Other Comments Or Suggestions:**

* page 6, line 303, LLM inference time being polynomial in the input length: Specific soft-quadratic time and linear space bounds are known for transformers, e.g., The Expressive Power of Transformers with Chain of Thought, ICLR-2024.

* page 2, line 70: the Karp citation is presumably for the seminal paper where Karp shows dozens of problems to be equally NP-complete. That work was done in 1972, not 2010!

* page 2, line 83: similarly, Cook's book was published earlier than 2011, around 2007 I believe. please double-check.

**Other Strengths And Weaknesses:**

N/A

**Questions For Authors:**

Please see some questions in the theoretical and empirical sections above.

**Relation To Broader Scientific Literature:**

To my limited knowledge of the area, the findings seem relevant, well-motivated, and novel. There are some concerns about clarity of the theoretical formalism and coverage of empirical findings, as discussed above.

**Theoretical Claims:**

The way the authors formulate and pitch the notion of "Quasi-Karp Equivalence" seems to lack some precision (unless I misunderstood something). Overall, I like this formal way of defining when two problems are equivalent. However, the following is causing confusion:

* While a reduction is naturally a one-way map (e.g., every "simpler" problem in the class NP reduces to an NP-complete problem, but not vice versa), I was naturally expecting *equivalence* to be a symmetric property. I am unclear why the authors didn't define problem formulations alpha and alpha' to be equivalent if alpha quasi-Karp reduce to alpha' AND alpha' quasi-Karp reduces to alpha. In fact, this is precisely how Karp equivalence is defined (two NP-complete problems are equivalent because they both reduce to each other).

* I don't quite see how the 2nd and 3rd bullet are different. Isn't it the case that f found be polynomial if and only if (some well-chosen) A computing f runs in polynomial time?

* In several places, the paper highlights the new notion being *instance-specific*. However, I really don't see how this is the case, after reading the description in section 3.2 and even after reading the detailed prompt in Appendix A. It seems to me a map from one problem formulation to another problem formulation. Just like in standard Karp reductions, it's a map from variables of one formulation to variables of another formulation. What is instance-specific about it? Wouldn't the transformation function be identical two different instances of, say, a traveling salesman problem? (I don't think this takes away anything from the findings, it's just confusing. E.g., the sentence "our method aims to evaluation the equivalence.... for a given instantiation of the problem" is confusing and doesn't seem to correctly represent what's being done)

* In several places, the paper highlights the new notion is a mapping from *solutions* to solutions, suggesting that a Karp reduction is not. But this isn't quite accurate --- a standard Karp mapping is precisely from an instance of a problem (e.g., network flow) to an instance of another problem (e.g., 3-SAT). It's always a mapping from variables of one problem to variables of the other problem, exactly like the proposed quasi-Karp mapping is. The only relevant difference is that the two problems considered are the same in this case, it's just that one has two distinct MILP formulations for it. But the mapping is still from variables of one to variables of the other.

---

> ### Author Rebuttal · Authors · 2025-04-01
>
> We are grateful for the reviewer's recognition of both the conceptual novelty of quasi-Karp equivalence and the empirical strength of EquivaMap on the benchmarked transformations. Below, we address the reviewer’s questions individually.
>
> ---
>
> >**C1**: “It's not clear whether EquivaMap will also work equally well in natural and real settings, … or in scenarios where multiple transformations are stacked together.”
>
> **A1**: We appreciate the reviewer’s concern about the performance of EquivaMap on compositional or stacked transformations. To assess this, we designed a new set of experiments where we **combined three transformation types**—Add Slack Variables, Add Valid Inequalities, and Replace by Linear Combinations. In this case, we have 59 LPs + 115 MILPs. We found that EquivaMap achieved **100% accuracy** on these new experiments with compositional transformations.  We will add these experimental results along with other stacked transformation experiments to our paper.
>
> >**C2**: "... I am unclear why the authors didn't define problem formulations alpha and alpha' to be equivalent if alpha quasi-Karp reduces to alpha' AND alpha' quasi-Karp reduces to alpha."
>
> **A2**: We appreciate the reviewer’s thoughtful observation! In classical complexity theory, equivalence is defined via a two-way reducibility. However, in our setting, we intentionally use a quasi-Karp (one-way) definition to better reflect the practical considerations of optimization modeling. As discussed in Section 3.2 (page 5), we relax the condition that a no-instance (which, in our setting, corresponds to an infeasible or suboptimal solution)
> under one formulation needs to be mapped to a no-instance of the other. This distinction is important in settings where a MILP formulation may exclude some, but not all, optimal solutions to improve efficiency. A common example is the addition of **symmetry-breaking constraints**, which preserve the objective and feasible set semantics, but eliminate functionally equivalent solutions. Because such transformations are often introduced in practice, we believe that a one-way reduction is sufficient and more appropriate for capturing formulation equivalence in applied contexts. Hence the use of the “quasi-” prefix.
>
> >**C3**: "I don't quite see how the 2nd and 3rd bullet are different. Isn't it the case that f is found to be polynomial if and only if A computing f runs in polynomial time?"
>
> **A3**: Thank you for this careful observation! While it’s true that in many cases these two conditions collapse, we intentionally separate them to illustrate the difference: it is theoretically possible for $A$ to output a description of $f$ in polynomial time (e.g., a program that implements $f$), but for $f$ itself to take super-polynomial time to evaluate. For example, $A$ could construct a branch-and-bound solver as $f$—in which case $A$ runs in polynomial time, but $f$ may not.
>
> >**C4**: "In several places, the paper highlights the new notion being instance-specific… What is instance-specific about it? Wouldn't the transformation function be identical to two different instances of, say, a traveling salesman problem?"
>
> **A4**: As the reviewer has mentioned, the mapping $f$ discovered by EquivaMap is **not instance-specific**. We feed the LLM a LaTeX-style formulation, where parameters such as  ```TotalPeople``` or  ```CapacityLargeUnit``` are left symbolic. The LLM then infers a symbolic mapping **between the two formulations**.
>
> However, the verification step in our algorithm is **instance-specific**: to check whether a proposed mapping preserves optimality, we instantiate the symbolic formulation with **concrete parameter values** and verify that the mapped solution is valid and optimal.
>
> To conclude, **the mapping is over formulations, while the verification check is over instances**.
>
> > **C5**: “In several places, the paper highlights the new notion is a mapping from solutions to solutions, … But the mapping is still from variables of one to variables of the other.”
>
> A5: We agree with the reviewer's comment and will make it clear in our paper that we are not “suggesting  that a Karp reduction is not.”
>
> ---
>
> We would also sincerely thank the reviewer for the other comments or suggestions session. We will update our citations to address these comments in the future version of the paper. We greatly appreciate your close reading and thoughtful suggestions, which help us improve both the precision and presentation of our work.

---

> > ### Comment · Reviewer_FhjQ · 2025-04-04
> >
> > Thank you for your explanations. They helped me better understand various choices from your perspective. Some, like using "quasi" to indicate that the equivalence is actually a one-sided reduction, still seem suboptimal to me, but I see your reasoning and you might also want to give it one more thought. In any case, it would be very valuable for many readers if you could include these clarifications in the paper.

---

> > > ### Author Response · Authors · 2025-04-07
> > >
> > > We sincerely thank the reviewer for the thoughtful feedback and the close reading of the paper. We are revising the draft to incorporate these clarification points and design details so that they are clearer to the broader audience.

---

### Official Review · Reviewer_46ti · 2025-03-18

**Overall Recommendation:** 4

**Summary:**

The paper introduces EquivaMap, an LLM-based framework for automatically verifying equivalence between combinatorial optimization formulations (MILP). It defines a new theoretical criterion called quasi-Karp equivalence, enabling robust equivalence checks even under transformations like variable scaling or auxiliary variables. Additionally, the authors create EquivaFormulation, a benchmark dataset of equivalent formulations. Experiments show EquivaMap achieves high accuracy, significantly outperforming baseline methods.

**Claims And Evidence:**

The claims appear reasonably supported by the designed experiments.

**Essential References Not Discussed:**

N/A

**Experimental Designs Or Analyses:**

I appreciate the current design of the experiments. The only extra experiment/discussion I would love to see is the running time of the EquivaMap algorithm: since the algorithm heavily relies on a capable solver to establish the robust equivalence mapping between problem formulations, the running time of such solvers and its relationship with the size of input formula should be taken into account.

**Methods And Evaluation Criteria:**

Yes, both the proposed quasi-Karp equivalence and the designed dataset look reasonable to me.

**Other Comments Or Suggestions:**

N/A

**Other Strengths And Weaknesses:**

The paper is well-written, with clear mathematical definitions and illustrative examples that effectively convey the key ideas.

**Questions For Authors:**

N/A

**Relation To Broader Scientific Literature:**

I envision this paper could be beneficial to the operations research.

**Theoretical Claims:**

There isn't any proof. The definitions look good to me.

---

> ### Author Rebuttal · Authors · 2025-04-01
>
> We sincerely thank the reviewer for the positive and thoughtful review, and for appreciating the design of our experiments. We also appreciate your suggestion to include a runtime discussion of the EquivaMap algorithm.
>
> Compared to execution accuracy, EquivaMap does not require additional MILP solving beyond the original problem instances. In particular, during the final verification step, we only evaluate whether the solution induced by the mapped variables satisfies the constraints in the target formulation and matches the known optimal objective value. This check is computationally lightweight and avoids any further calls to optimization solvers. In other words, once the MILPs are solved initially, **no further solving is needed** to verify a candidate mapping.
>
> In practice, EquivaMap typically completes in **a few seconds per instance**, including both the LLM inference step and the verification step (typically very fast). We will add a more detailed runtime analysis in the future version of the paper.

---

### Official Review · Reviewer_NwSG · 2025-03-18

**Overall Recommendation:** 4

**Summary:**

This paper discusses how LLMs (+ NLP) can help with automatic checking of equivalences in the context of combinatorial optimization reductions.

**Claims And Evidence:**

Yes, the experiments are convincing.

**Essential References Not Discussed:**

No.

**Experimental Designs Or Analyses:**

Yes, the ones in the main body.

**Methods And Evaluation Criteria:**

Yes.

**Other Comments Or Suggestions:**

None.

**Other Strengths And Weaknesses:**

Strengths:
The paper gives a novel connection between computational complexity and LLMs.
En route, the authors introduce a new kind of reductions, namely quasi-Karp reductions, which might be of independent interest.

Weaknesses:
Technically, the paper is not so deep :)

**Questions For Authors:**

Page 4:
Since you are mentioning cutting planes, do you see any connections of your work to proof complexity, etc.?

Page 5:
Algorithm 1:
I think you should elaborate on the LLM prompt.
This step is a bit hand-wavy :)

Page 6:
Why is it a reasonable assumption that LLM inference is in poly-time?

Page 8:
I think that in the Discussion, you should elaborate more on future work directions.

**Relation To Broader Scientific Literature:**

They explore a novel direction in the use of LLMs, whereby LLMs are used as a copilot in checking equivalence between instances that are inputs/outputs of reductions.

**Theoretical Claims:**

N/A.

---

> ### Author Rebuttal · Authors · 2025-04-01
>
> We sincerely thank the reviewer for their positive and thoughtful comments, and for recognizing the novelty of our work:
>
> > “The paper gives a novel connection between computational complexity and LLMs.”
>
> Below, we address the specific questions and suggestions:
>
> ---
>
> > **Q1**: "Since you are mentioning cutting planes, do you see any connections of your work to proof complexity, etc.?"
>
> **A1**: We indeed view our work as a first step toward automated reasoning about proof complexity. If Quasi-Karp Equivalence can be applied to two instances of two different families of optimization problems, it might be viewed as a potential **heuristic** for checking if two optimization problems are of the same complexity. That said, we recognize that these connections to proof complexity remain speculative and require further investigation. We thank the reviewer for highlighting this promising line of inquiry.
>
> > **Q2**: "Algorithm 1: I think you should elaborate on the LLM prompt. This step is a bit hand-wavy"
>
> **A2**: Thank you for this suggestion! Currently,  in Appendix A, we provide the exact prompt template generator used in all of our current experiments. We will include more details on this in our paper.
>
> > **Q3**: "Why is it a reasonable assumption that LLM inference is in poly-time?"
>
> **A3**: Thank you for raising this question. As noted in The Expressive Power of Transformers with Chain of Thought (ICLR 2024) as well as comments by Reviewer FhjQ,   it is shown that “a polynomial number of steps [in terms of the input] turns transformers into strong reasoners.”
>
>
>
> Moreover, in EquivaMap, we typically call the LLM **once per set of variables**, not per individual variable. Since the number of sets of variables is often **orders of magnitude smaller** than the total number of variables in an instance, this further supports the practicality and tractability of our approach. We will add these discussions in the paper.
>
> > **Q4**: "I think that in the Discussion, you should elaborate more on future work directions."
>
> **A4**: We agree that the future directions are rich in this space and worth expanding. In the final version, we will include more concrete directions, such as the potential usage of quasi-Karp equivalence to work as a heuristic for checking if two optimization problems are of the same complexity.

---

### Decision · Program_Chairs · 2025-05-01

**Decision:**

Accept (poster)

**Comment:**

This paper focuses on the problem of of automatically verifying equivalence between combinatorial optimization formulations (MILP) via LLMs and to this end, proposes the notion of quasi-karp like reductions. The reviewers found the paper's claims to be well motivated and supported: The writing at times appears imprecise and there is detailed feedback by reviewers. The reviewers who have provided feedback have deep expertise in computational complexity and therefore, I strongly recommend authors improve the final version by taking into the feedback of reviewers into account else authors risk the paper to be ignored by theoretical community.